# Characterization of Electrical Heating Performance of CFDM 3D-Printed Graphene/Polylactic Acid (PLA) Horseshoe Pattern with Different 3D Printing Directions

**DOI:** 10.3390/polym12122955

**Published:** 2020-12-10

**Authors:** Hyelim Kim, Sunhee Lee

**Affiliations:** 1Research Institute of Convergence Design, Dong-A University, Busan 49315, Korea; hyelim1221@gmail.com; 2Department of Fashion Design, Dong-A University, Busan 49315, Korea

**Keywords:** conveyor-fused deposition modeling 3D printer, graphene/PLA, horseshoe pattern, 3D printing direction, electrical heating property

## Abstract

This study manufactured a horseshoe pattern (HP)-type electrical heating element based on a graphene/polylactic acid (GR/PLA) filament using CFDM (conveyor-fused deposition modeling) 3D printing technology, which is a new manufacturing process technology. CFDM 3D printing HP was fabricated in the different printing directions of 0°, 45°, and 90°. To confirm the effects of different 3D printing directions, the morphology, surface resistivity, and electrical heating properties of the different HPs were analyzed. In addition, the CFDM 3D-printed HPs made using different printing directions were printed on cotton fabric to confirm their applicability as fabric heating elements, and their electrical heating properties were measured. Regarding the morphology of the GR/PLA-HP, each sample was stacked according to the printing direction. It was also confirmed through FE-SEM images that the graphene was arranged according to the printing direction in which the nozzle moved. In the XRD pattern analysis, the GR/PLA-HP samples showed two diffraction peaks of PLA and graphene. The sizes of those peaks were increased in the order of 90° < 45° ≤ 0° according to the printing direction, which also affected the electrical and electric heating properties. The surface resistivities of the GR/PLA-HP samples were shown to be increased in the order of 0° < 45° < 90°, indicating that the electrical properties of GR/PLA HP printed at 0° were improved compared to those of the other samples. When 30 V was applied to three GR/PLA-HP samples according to the printing direction, the surface temperatures were decreased in the order of 0° < 45° < 90°, and the samples were indicated as 83.6, 80.6, and 52.5 °C, respectively; the same result was shown when the samples were printed on cotton fabric. Therefore, it was confirmed that the GR/PLA CFDM 3D-printed HP sample printed at 0° direction showed low surface resistivity and high surface temperature, so that improving the electrical heating properties.

## 1. Introduction

3D printing technology is applied in many industrial fields, including the production processes of buildings, aerospace, biomaterials, protective structures, and textiles [1]. Among them, the FDM (fused deposition modeling) 3D printing process is used in many applications due to its ease of use in producing desired samples [2]. Several studies [3,4] have investigated the use of FDM 3D printing technology in fabricating textiles. However, textiles are difficult to manufacture in large quantities through this method due to the limitations of the bed size and the nozzle movement. Recently, to solve the problem, conveyor-fused deposition modeling (CFDM) 3D printing process method was developed. The CFDM 3D printing technology can produce long and series printings, so it has the advantage of being suitable for mass production because it can 3D print infinitely on the Z-axis [5].

Generally, thermoplastic polymers filaments such as polylactic acid (PLA), acrylonitrile butadiene styrene (ABS), and thermoplastic polyurethane are mainly used in FDM 3D printing [1]. To impart conductivity to the thermoplastic polymer-based filament, conductive filaments are manufactured by adding carbon-based nanofillers such as carbon black (CB), carbon nanotube (CNT), and graphene (GR) [6]. Many studies are also using FDM 3D printing to make flexible conductive circuits, electrodes, and sensors for wearable devices [7,8,9,10]. Further, several studies report that using carbon nanofillers in polymer matrix affects their alignment [11,12]. In addition, previous studies on 3D printing using conductive filaments have reported that the 3D printing direction and the raster angle influence the electrical properties of the final product [13,14,15]. Chen et al. [11] report that the significant peak of the carbon-based nanofiller attributed to (002) plane could show increased peak intensity size and improved electrical properties due to the X-ray scanning direction facing the nanofiller more. Watschke et al. [14] have studied the electrical resistance of 3D-printed sample with three-types of conductive filaments with varying raster angles; the raster angles were controlled among 0°, ±45°, and 90°. The conductive 3D-printed samples printed with three types of filaments showed a tendency of increasing electrical resistance in the order of 0° < ±45° < 90°. It was reported that the conductivity was increased by the high orientation of the fillers in the extrusion direction.

Most studies using FDM 3D printing technology with carbon nano-based filament have mainly focused on conductive circuits or sensors, and there is insufficient research on the application of this filament as an electrical heating element. In addition, most fabric heating elements are produced through various coating methods using conductive ink [16,17,18,19]. Thus, this present study aims to manufacture and analyze horseshoe pattern (HP)-type heating elements and fabric heating elements using CFDM 3D printing-based graphene (GR)/PLA filament with different 3D printing directions. The specific objects produced are as follows. First, the HP-type electrical heating element was manufactured using CFDM 3D printing-based GR/PLA filament with designing and modeling for each printing direction of 0°, 45°, and 90°. Then, to confirm the changes in the properties with the different 3D printing directions, the samples were analyzed in terms of their morphology, XRD pattern, electrical properties, and electrical heating properties. Finally, to confirm its applicability as a fabric heating element, CFDM 3D-printed HP/cotton composite fabric was prepared using GR/PLA with different 3D printing directions, and the electrical heating properties of each sample were investigated.

## 2. Materials and Methods

### 2.1. Materials

In this study, PLA (TPC Mechatronics, Incheon, Korea) and graphene (GR)/PLA (Blackmagic 3D Ltd., Ronkonkoma, NY, USA) filament were selected for 3D printing. These filaments are FDM technology-based 3D printing materials, each with a diameter of 1.75 mm. The filaments were stored in a desiccator at a standard condition before being used for analysis and 3D printing.

### 2.2. Preparation of CFDM 3D-Printed Horseshoe Pattern

#### 2.2.1. Design and 3D Modeling of Horseshoe Pattern

Figure 1 shows the scheme of designing the horseshoe pattern (HP). To prepare a 3D-printed HP sample, one line of HP was first designed with 50.0 mm × 6.5 mm of a 2.5 mm line width, according to our previous study [20]. Next, a final modeling with a size of 50.0 mm × 50.0 mm × 1.0 mm was created and converted to an STL file using the 123D design program. Finally, it was transformed into the g-code file with three different printing directions of 0°, 45°, and 90° to prepare for CFDM 3D printing using the Blackbelt Cura program (Figure 2).

#### 2.2.2. CFDM 3D Printing Conditions

The 3D printing in this study was performed using a conveyor-type FDM (CFDM) 3D printer (Blackbelt 3D B.V., Belfeld, The Netherlands). We controlled the 3D printing conditions such as the temperatures of the nozzle and the bed, the printing speed, and the infill according to filament type, according to our previous study [5]. The 3D printing conditions of both the PLA and GR/PLA filaments are summarized in Table 1, and the sample codes of the CFDM 3D-printed horseshoe pattern with various 3D printing conditions are listed in Table 2.

### 2.3. Preparation of CFDM 3D-Printed Horseshoe Pattern/Cotton Composite Fabric

To confirm the applicability of the CFDM 3D-printed horseshoe pattern as a fabric heating element, the next step was to manufacture the CFDM 3D-printed horseshoe pattern on cotton fabric using GR/PLA filament. The untreated fabric used in the experiment was a cotton fabric with a thickness of 0.28 ± 0.01 mm, a weight per unit area of 0.011 ± 0.001 g/cm^2^, and warp and weft yarns with the same density of 70/inch. Untreated cotton fabric was prepared in the dimensions of 60 mm × 60 mm. Three types of 3D modeling of the horseshoe pattern by 3D printing direction were printed on the untreated cotton fabric with a thickness of 1 mm using a CFDM 3D printer. Table 3 presents the sample codes and digital images for the three types of CFDM 3D-printed horseshoe pattern/cotton composite fabric.

### 2.4. Characterization

#### 2.4.1. Morphology

To analyze the 3D-printed horseshoe pattern with different 3D printing directions, the morphology of the 3D-printed sample’s surface and lateral were measured through fabric image analysis microscopy (NT 100, Nextec, Gunpo, Korea) at ×6.5 magnification and FE-SEM (JSM-6700F, Jeol., Tokyo, Japan) at magnifications of ×300, ×1000, ×10,000, and ×15,000.

#### 2.4.2. X-ray Diffraction (XRD)

X-ray diffraction (XRD) spectra of the 3D-printed horseshoe pattern structure made with different 3D printing directions were taken using an X-ray diffractometer (Ultima IV, Rigaku Co., Ltd., Tokyo, Japan) with Ni-filtered CuKα radiation. The analysis was conducted under a measuring range of 5–65° and a scanning speed of 1°/min. At this time, the X-ray beam direction was a sample printed at 0°, and XRD was measured each direction of sample based on this.

#### 2.4.3. Surface Resistivity

To measure the electrical properties of the samples, the surface resistivity of each line of the 3D-printed samples made with different 3D printing directions was measured with a multimeter (ST850A, Saehan Tester, Co., Ltd., Busan, Korea) based on the AATCC-76 method. The surface resistivity (R_s_) was calculated using Equation (1):R_s_(Ω/sq) = (W/D) × R(1)
where R is the resistance (Ω) measured by the multimeter, W is the width (mm) of the sample, and D (mm) is the distance between the two electrodes.

#### 2.4.4. Electrical Heating Property

The electrical heating property of each line of the CFDM 3D-printed HP samples and the composite cotton fabrics made with different 3D printing directions was investigated based on surface temperature with various voltages while using a DC power supply (CPS-2450B, CHUNGPA EMT Co., Ltd., Bucheon, Korea). Alligator clips were connected to both sides of each sample, and voltage was applied from 5 to 30 V with 5 V (DC) intervals for 3 min; a thermal imaging camera (FLIR i5, FLIR Systems Inc., Wilsonville, OR, USA) was used to measure the surface temperature after applying these voltages.

## 3. Results and Discussion

### 3.1. Morphology of CFDM 3D-Printed Horseshoe Pattern Manufactured with Different 3D Printing Directions

Table 4 presents the morphologies of the CFDM 3D-printed horseshoe patterns made using PLA and GR/PLA filament with different 3D printing directions. Examining the digital images of the PLA-HP and GR/PLA-HP 3D-printed samples confirmed that both samples could be printed in the three 3D printing directions of 0°, 45° and 90°. At this time, in the case of the GR/PLA sample, the distance between the stacked layers according to the 3D printing direction was studied. It was found that the samples output at 0° are stacked without showing a gap, whereas the samples output at 45° and 90° show that some gaps are formed between the layers. Regarding the morphologies of the PLA-HP and GR/PLA-HP samples according to the 3D printing direction, it was found that each sample was stacked by the printing directions at 0°, 45°, and 90° on the surface of the sample, and in the GR/PLA samples, it was confirmed that the degree of gap of the stacked layer was changed according to the stacking direction. Regarding the lateral parts of the samples, the used CFDM 3D printer has a gantry angle, which makes it possible to control the angle, so the samples were printed out at 15° and showed a tilt according to the gantry angle.

To confirm stacked direction of layer and arrangement of graphene in the 3D-printed horseshoe pattern samples, FE-SEM images were taken at the surface and the lateral parts. Table 5 shows the surface and lateral FE-SEM images of the graphene/PLA-based CFDM 3D-printed horseshoe pattern made with different 3D printing directions. As shown in the surface image of the sample at a magnification of ×10,000 or less, each printing direction, which is the moving direction of the nozzle, was found in the surface of the sample, thus confirming that the layers were stacked. Additionally, in the CFDM 3D-printed HP samples of the surface at ×15,000 magnification, it seems that the arrangement of graphene in the GR/PLA-HP samples is also affected because the GR/PLA filaments stack layers according to the nozzle movement direction. In case of the lateral SEM images of 3D-printed sample, it was also confirmed that the stacked layer depended on the 3D printing direction at the lateral of the 3D-printed samples. As shown in the lateral image of the sample at a magnification of ×300, the molten GR/PLA filament was stacked according to the nozzle movement direction. As mentioned in the surface morphology, GR/PLA-HP samples were affected by the gantry angle, and GR/PLA-HP00 was straight, whereas GR/PLA-HP45 and GR/PLA-HP90 show slopes. In addition, it was confirmed that the contact area between the layer-by-layer of the GR/PLA-HP00 sample was wider than that of GR/PLA-HP45 and GR/PLA-HP90. In addition, at the ×1000 and ×10,000 magnifications, it represented the one layer of the GR/PLA-HP samples. At that time, the printing tracks were exhibited, and it was also shown that the layers of each sample were affected by the printing direction, which is the nozzle movement direction, and graphene in the molten GR/PLA filament seems to be arranged accordingly. Therefore, it is confirmed that when the GR/PLA filament is melted through the nozzle and is placed on the conveyor belt type bed along the printing direction, the contact area of layer-by-layer of each sample and the arrangement of graphene particles are also influenced by the 3D printing direction, which is the direction of the nozzle movement.

### 3.2. XRD Analysis of CFDM 3D-Printed Horseshoe Pattern Manufactured with Different 3D Printing Directions

To confirm the effect of the 3D printing direction on the highly oriented structure of the PLA and GR/PLA 3D-printed HP samples, XRD patterns were taken and analyzed as shown in Figure 3. The planes for XRD testing of samples by different 3D printing directions were tested parallel, diagonal, and perpendicular to the orientation direction. Regarding the results of the XRD patterns of the PLA and GR/PLA 3D-printed HP samples made with different 3D printing directions, the diffraction patterns of the PLA 3D-printed HP samples at 0°, 45°, and 90° presented broad peaks at 2θ = 17.1°, 15.7°, and 17.3°, respectively. In the case of GR/PLA 3D-printed HP samples made with different 3D printing directions, two of the dominant peaks were indicated. The peaks at around 17.5° and 27.4° were identified as PLA and graphene peaks, respectively. Regarding the PLA 3D-printed HP samples, the intensity was similar according to the three different 3D printing directions. On the other hand, in the GR/PLA 3D-printed HPs, each peak was observed to have a different intensity according to the 3D printing direction. The peak intensity sizes were similar when the 3D printing directions were 0° and 45°, but the peak intensity of the sample 3D printed at 90° showed a sharp decrease compared to those of the GR/PLA 3D-printed HP samples at 0° and 45°. Previous studies [11,20] reported that the intensity of the diffraction signal originating from the carbon-based nanofiller/polymer composite is affected by the alignment of the carbon-based nanofiller. They reported that the significant peak of carbon-based nanofiller attributed to the (002) plane could show an increased peak intensity size due to the X-ray scanning direction facing the nanofiller more. Haney et al. [20] have fabricated graphene nanoplatelets for 3D printing and studied printability and performance of their 3D printing structures. They have compared the intensity of the (002) peak at 2θ = 26.4°, which is the peak of the graphene nanoplatelets, and the results have reported that the peak intensity of an in-plane alignment, which is the parallel to print direction, is higher than an out of plane alignment, which is the transverse to print direction. They also have been referred that it is because the shear force generated during the deposition of the composite, resulting in a shift in the orientation of the 2D platelet-like fillers and that is induced on the filler during printing, causes these nanoparticles to be oriented parallel to the direction of the print [20]. Therefore, it was confirmed that the graphene particles in the 3D-printed samples were arranged according to the 3D printing direction due to the shear stresses during the deposition from nozzle to bed, and this was found to affect the highly oriented structure. In addition, the contact area of layer-by-layer of samples was also attributed to peak intensity. Accordingly, it was found that the GR/PLA-HP00 that was identical to the X-ray scanning direction and possessed wider contact area of layer-by-layer among GR/PLA-HP samples represented the highly oriented structure.

### 3.3. Surface Resistivity of CFDM 3D-Printed Horseshoe Pattern Manufactured with Different 3D Printing Directions

To investigate the changes in the electrical properties of CFDM 3D-printed HP samples according to different 3D printing directions, the surface resistivity of each line of the PLA-HP and GR/PLA-HP samples was analyzed, as shown in Figure 4. As mentioned previously, carbon nanomaterials have different electrical properties depending on their orientation [11]. In addition, the 3D printing directions and raster angles of the samples are determined by the direction in which the nozzle moves in the 3D printing process [5,13,14,15,20]. In this study, the electrical surface resistivities of the PLA-HP samples made with different 3D printing directions were all indicated to be higher than 10^13^ Ω/sq, which is confirmed by the insulating properties of PLA. On the other hand, in the case of the GR/PLA-HP samples made with different 3D printing directions, the surface resistivity values of GR/PLA-HP00, GR/PLA-HP45, and GR/PLA-HP90 were 1.89 × 10^3^ ± 3.1 × 10^2^, 2.02 × 10^3^ ± 3.3 × 10^2^, and 3.17 × 10^3^ ± 7.3 × 10^2^ Ω/sq, respectively. Thus, the surface resistivity was increased in the order of GR/PLA-HP00 < GR/PLA-HP45 < GR/PLA-HP90. Chen et al. [11] reported that an increase in crystallinity could affect the electrical properties due to a corresponding increase in the alignment of graphene particles. Watschke et al. [14] reported the electrical properties of 3D printing samples made with three different raster angles. The results of this study indicate that the increase in electrical conductivity is greater when the raster angle of 0° is used. This is attributed to the increase in conductivity caused by the high orientation of fillers in the extrusion direction. Haney et al. [20] analyzed the effect of conductivity and orientations of graphene nanoplatelets composite. The results of the electrical conductivity were improved at parallel to print direction than transverse to print direction. They reported that the shear stresses generated during the deposition of the composite resulted in a shift in the orientation of the graphene nanoplatelets fillers and the ability to tune the electrical properties of a printed structure with a constant loading of filler. In addition, it is confirmed that these properties are related to the contact area of the stacked layer-by-layer. Zhang et al. [21] prepared an FDM 3D-printed sample with ABS/carbon black composite and analyzed the resistivity and its anisotropy characterization according to the process parameters and at both vertical and horizontal directions.

The resistivity in the sample of vertical direction was approximately 3.83 times that of horizontal direction, and they suggested that the interlayer bonding, including the bonding area and the bonding quality, may significantly influence the resistivity in the vertical direction of the printed parts. In our present study, as shown in the analysis of morphology, the layer-by-layer of GR/PLA-HP00 sample was stacked horizontally. However, GR/PLA-HP45 and GR/PLA-HP90 were stacked at diagonal and vertical direction, thus, the contact or bonding area was in order of GR/PLA-HP90 < GR/PLA-HP45 < GR/PLA-HP00, and the contact or bonding area of GR/PLA-HP00 could be more wider than GR/PLA-HP45 and GR/PLA-HP90; it created more conductive paths. It was shown that the contact or bonding area also could be influenced by the surface resistivity. Thus, it was found that the surface resistivity of GR/PLA-HP00, which 3D-printed with horizontal direction, was lower than those of the other samples that caused more contact resistance due to printing at diagonal and vertical direction because more conductive paths could be created due to the wider bonding area.

### 3.4. Electrical Heating Properties of CFDM 3D-Printed Horseshoe Pattern Manufactured with Different 3D Printing Directions

To analyze the electrical heating properties of PLA and GR/PLA 3D-printed horseshoe pattern samples made with different 3D printing directions, a thermal image of each of the lines of the samples was taken when different voltages ranging from 5 to 30 V were applied, and these are presented in Table 6 and Table 7. As shown in Table 6, it was confirmed that the PLA-HP samples made with different 3D printing directions exhibited no variations in their electrical heating properties with different applied voltages, which is attributed to the insulation property of PLA. However, as shown in Table 7, the surface temperatures of the GR/PLA-HP samples increase as the applied voltage is increased. At this time, all GR/PLA-HP samples express a surface temperature of 60 °C or more and exhibit the characteristic of being made flexible by heat. It is confirmed that the PLA exhibits low glass transition temperatures of around 59 °C, which affects the GR/PLA 3D-printed HP samples [5]. Figure 5 indicates the surface temperatures of the CFDM 3D-printed horseshoe pattern made with different 3D printing directions. In the case of GR/PLA-HP00 and GR/PLA-HP45, the surface temperature was increased more rapidly than it was for GR/PLA-HP90. With the applied voltage of 30 V, the surface temperatures of GR/PLA-HP00, GR/PLA-HP45, and GR/PLA-HP90 were indicated to be 83.6 ± 9.2, 80.6 ± 5.8, and 52.5 ± 3.1 °C, respectively. It is confirmed that the orientation of the graphene particles arranged by the 3D printing direction, as confirmed above, improves the crystallinity and electrical properties, and thus affects the electrical heating properties. In general, the power of the heating generated by an electrical heating element is proportional to the samples of its resistance and the square of the current [16]. Therefore, it was found that GR/PLA-HP00 and GR PLA-HP45, which had higher currents than GR/PLA-HP90, exhibited excellent electrical heating properties.

To confirm the current–voltage (I–V) relationship of 3D printing directions of CFDM 3D-printed horseshoe pattern samples made with GR/PLA filament, I–V curve of GR/PLA-HP samples are shown in Figure 6. As shown in Figure 6, the currents of GR/PLA-HP00, GR/PLA-HP45, and GR/PLA-HP90 showed a tendency to gradually increase. When the voltage was applied from 5 to 30 V, the current values of GR/PLA-HP00, GR/PLA-HP45, and GR/PLA-HP90 were shown to range from 0.00 to 0.04 A, from 0.00 to 0.04 A, and from 0.00 to 0.03 A, respectively. In addition, the GR/PLA-HP samples were influenced by the 3D printing direction, and the currents were increased in the order of GR/PLA-HP90 < GR/PLA-HP45 < GR/PLA-HP00 at the maximum voltage of 30 V. This tendency can be seen in the results of both surface resistivity and surface temperature. As mentioned above, it was confirmed that the crystallinity and electrical properties were changed according to the alignment of graphene in the graphene/polymer composite [11,14]. Since this tendency improves the electrical properties of GR/PLA-HP00, where the graphene alignment direction is the same as the X-ray beam direction, it is confirmed that its current can flow most easily among the three directions. It is generally known that the power of electric heating properties has a substantial influence on current, according to Joule’s law [16]. As a result, when the same applied voltage of 30 V was applied, GR/PLA-HP00, which had the largest current value, exhibited a high surface temperature and uniform distribution, and it was confirmed to have the best electrical heating properties among the samples tested.

### 3.5. Electrical Heating Properties of CFDM 3D-Printed Horseshoe Pattern Printed on Cotton Fabric

To investigate the possibility of the fabrication of a fabric heating element using CFDM 3D printing technologies, CFDM 3D-printed horseshoe pattern/cotton composite fabric was produced by printing along three different directions (Table 3). For this purpose, the CFDM 3D printing horseshoe pattern/cotton composite fabric was manufactured by 3D printing six horseshoe patterns with the size of 50 mm × 50 mm as GR/PLA onto a 60 mm × 60 mm substrate fabric. The electric heating properties were tested by the same method mentioned previously. The surface temperature was measured accordingly by applying a voltage from 5 to 30 V for 3 min.

Table 8 presents the IR images of the GR/PLA CFDM 3D-printed horseshoe pattern/cotton composite fabric made with different 3D printing directions. As shown in Table 8, for GR/PLA/Co-HP00, GR/PLA/Co-HP45, and GR/PLA/Co-HP90, the heat distribution is confirmed to expand as the applied voltage increases, and accordingly, the surface temperature tends to gradually increase. In general, in the case of the electric heating element made of a carbon nanomaterial, it can be seen as a phenomenon caused by disordering of the thermal motion when the direction of free electrons is offset [16]. As the voltage is applied to the GR/PLA/Co-HP samples prepared in this study, current flows in a region dispersed in a percolation state of graphene in the graphene/polymer matrix. At this time, when free electrons flow in the region where graphene is dispersed, it becomes disordered without direction, and accordingly, the heat distribution is expanded and the temperature appears to rise.

As shown in Figure 7a, the surface temperatures of GR/PLA/Co-HP00, GR/PLA/Co-HP45, and GR/PLA/Co-HP90 were 82.6 ± 8.7, 79.6 ± 7.8, and 61.5 ± 3.7 °C, respectively, when the voltage of 30 V was applied. These samples showed the same tendency as the GR/PLA-HP samples, and the surface temperature increased in the order of GR/PLA/Co-HP90 < GR/PLA/Co-HP45 < GR/PLA/Co-HP00. At this time, the GR/PLA/Co-HP samples were found to be more stable than the GR/PLA-HP samples, which indicated flexible characteristics at about 60 °C, which is the glass transition temperature of PLA [5]. It was confirmed that it showed thermal stability, which was attributed to the cotton fabric used as a substrate fabric. However, regarding the current, it was found that GR/PLA/Co-HP00, GR/PLA/Co-HP45, and GR/PLA/Co-HP90 showed similar values as the applied voltage increased (Figure 7b). The results showed that there was a difference in heat generation performance despite the fact that the same applied voltage and current were supplied, and this difference was attributed to the alignment of graphene according to the 3D printing direction. Therefore, it was confirmed that the applicability of the GR/PLA/Co-HP sample to the fabric heating element and the electrical heating properties of the GR/PLA/Co-HP00 sample were excellent.

## 4. Conclusions

This study was conducted to manufacture and analyze a horseshoe pattern type electrical heating element and test the applicability of the fabric heating element using a CFDM 3D printing-based graphene/PLA filament and made with different printing directions of 0°, 45°, and 90°. The changes in the morphologies of the PLA and GR/PLA HP samples according to the 3D printing direction confirmed that each sample was stacked by the printing direction at 0°, 45°, and 90°. As shown in the FE-SEM images, graphene was arranged according to the 3D printing direction, which is the direction in which the nozzle moved. To analyze the effect of 3D printing direction on the crystallinities of PLA and GR/PLA 3D-printed HP samples, XRD was conducted. The XRD patterns of the GR/PLA 3D-printed HP patterns showed two dominant peaks; these peaks at around 2θ = 17.5° and 2θ = 27.4° were identified as PLA and graphene peaks, respectively. In addition, in the GR/PLA 3D-printed HP samples, the peak intensity sizes of GR/PLA-HP00 and GR/PLA-HP45 were similar to each other, but that of GR/PLA-HP90 showed a sharp decrease relative to those of GR/PLA-HP00 and GR/PLA-HP45. This tendency affected the electrical properties. The surface resistivities of GR/PLA 3D-printed HP samples depending on the 3D printing directions showed an increase in the order of GR/PLA-HP00 < GR/PLA-HP45 < GR/PLA-HP90. This tendency was also shown for the electrical heating properties. GR/PLA-HP00 and GR/PLA-HP45 were found to be over 50 °C at the applied voltage of 25 V, whereas GR/PLA-HP90 was shown to be below 50 °C. When 30 V was applied to GR/PLA-HP00, GR/PLA-HP45, and GR/PLA-HP90, they were found to be 83.6, 80.6, and 52.5 °C, respectively. The same tendency was shown when applied to cotton fabrics. Thus, it was confirmed that the electrical resistivity and electrical heating properties of the GR/PLA 3D-printed HP samples were improved when 3D printed at the 0° direction, and they are confirmed to be suitable for fabric heating elements.

## Figures and Tables

**Figure 1 polymers-12-02955-f001:**
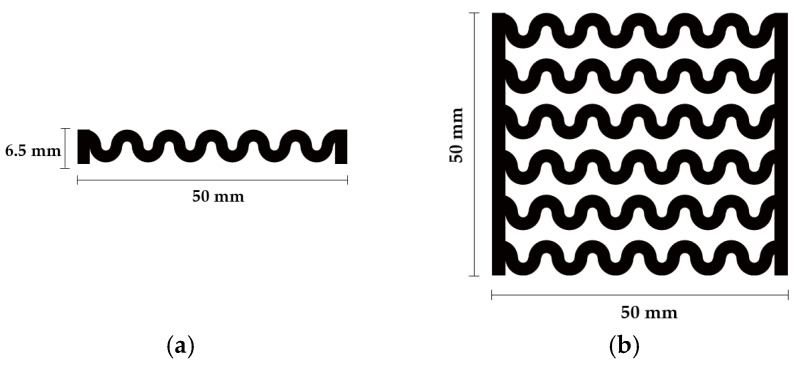
Scheme of designing (**a**) one line and (**b**) six lines of horseshoe pattern.

**Figure 2 polymers-12-02955-f002:**
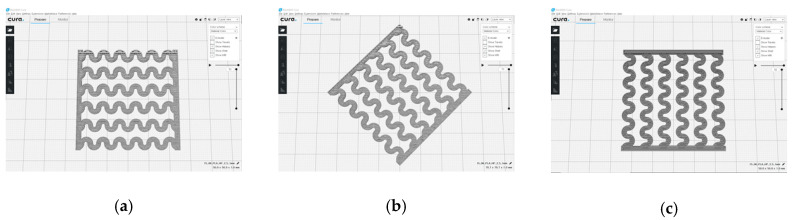
3D modeling of horseshoe patterns with different printing directions among (**a**) 0°, (**b**) 45°, and (**c**) 90°.

**Figure 3 polymers-12-02955-f003:**
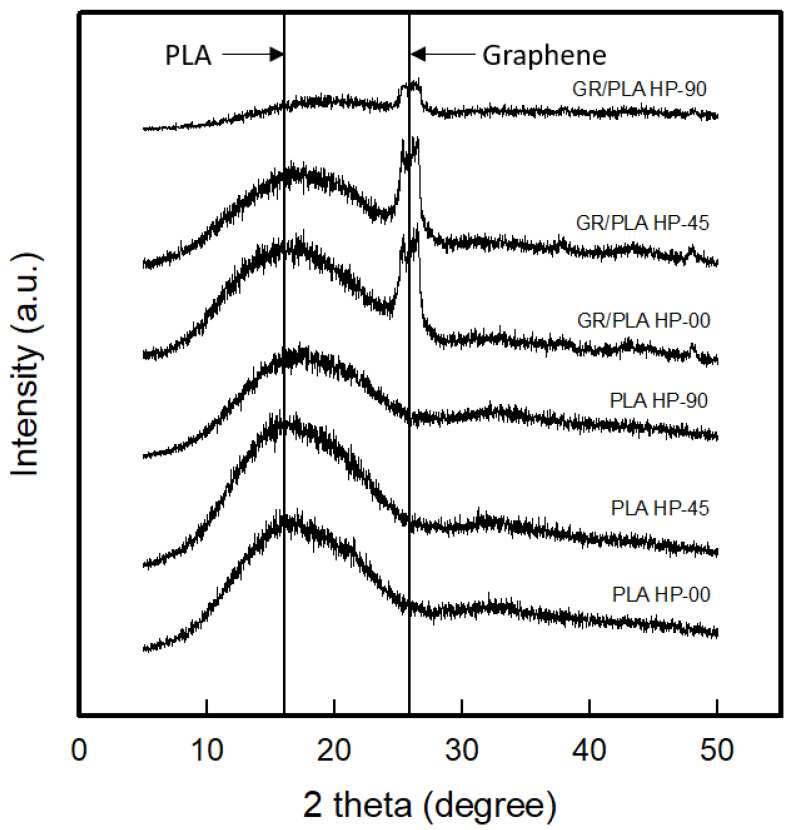
XRD patterns of conveyor-fused deposition modeling (CFDM) 3D-printed horseshoe pattern made with different 3D printing directions.

**Figure 4 polymers-12-02955-f004:**
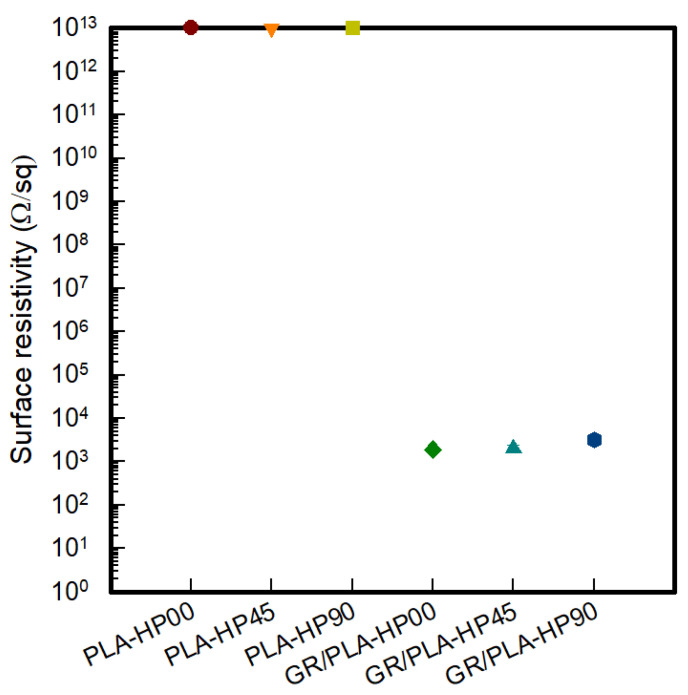
Surface resistivity of CFDM 3D-printed horseshoe pattern with different 3D printing directions.

**Figure 5 polymers-12-02955-f005:**
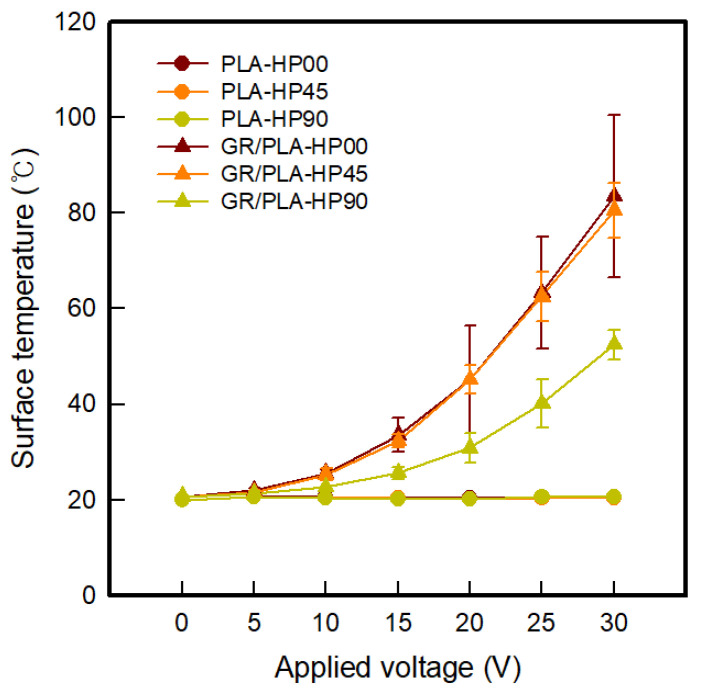
Surface temperatures of CFDM 3D-printed horseshoe pattern with different 3D printing directions.

**Figure 6 polymers-12-02955-f006:**
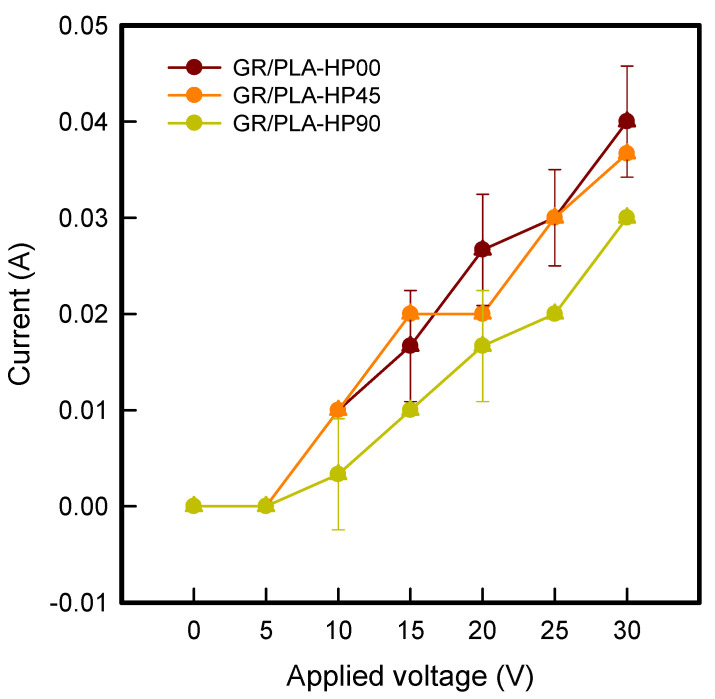
Current–voltage (I–V) of CFDM 3D-printed horseshoe pattern with different 3D printing directions.

**Figure 7 polymers-12-02955-f007:**
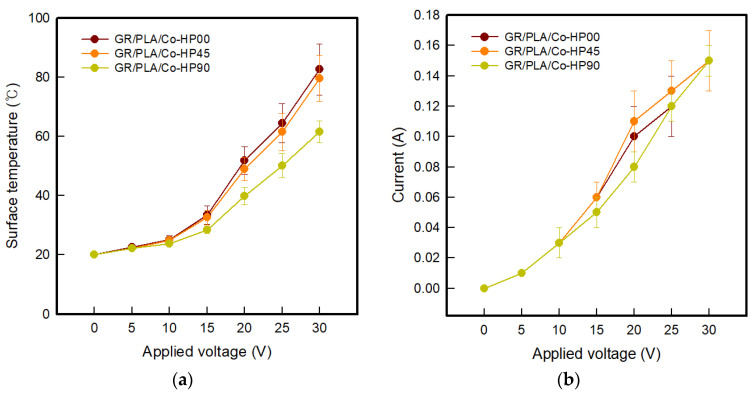
(**a**) Surface temperature-voltage and (**b**) current–voltage of CFDM 3D-printed horseshoe pattern cotton fabric with different 3D printing directions.

**Table 1 polymers-12-02955-t001:** Conveyor-fused deposition modeling (CFDM) 3D printing conditions used with polylactic acid (PLA) and graphene (GR)/PLA filaments in this study.

Parameters	PLA	GR/PLA
Belt offset (mm)	0.2	0.2
Layer height (mm)	1.8	1.8
Temperature of nozzle (°C)	215	220
Temperature of bed (°C)	75	50
Printing speed (mm/s)	40	30
Infill speed (mm/s)	30	22.5
Infill (%)	100	100
Gantry angle (°)	15	15

**Table 2 polymers-12-02955-t002:** Sample codes of CFDM 3D-printed horseshoe pattern with various 3D printing conditions.

3D Printing Direction	PLA		GR/PLA	
	Sample Code	Digital Image	Sample Code	Digital Image
0°	PLA-HP00	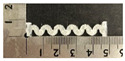	GR/PLA-HP00	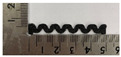
45°	PLA-HP45	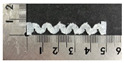	GR/PLA-HP45	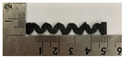
90°	PLA-HP90	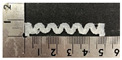	GR/PLA-HP90	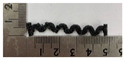

**Table 3 polymers-12-02955-t003:** Sample codes and digital images of CFDM 3D-printed horseshoe pattern/cotton composite fabric with different 3D printing directions.

3D Printing Direction	Sample Code	Digital Image
0°	GR/PLA/Co-HP00	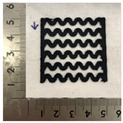
45°	GR/PLA/Co-HP45	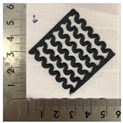
90°	GR/PLA/Co-HP90	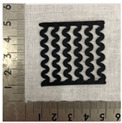

**Table 4 polymers-12-02955-t004:** Morphology of CFDM 3D-printed horseshoe pattern by different 3D printing direction.

Measured Part	Magnification	Sample Code		
		PLA-HP00	PLA-HP45	PLA-HP90
Digital image		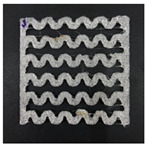	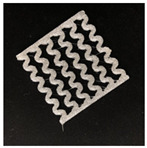	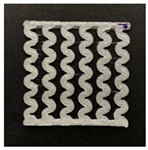
Surface	×6.5	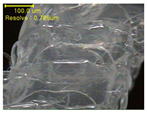	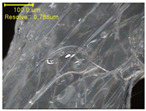	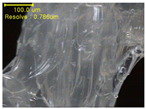
Lateral		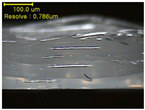	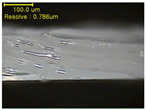	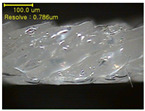
**Measured Part**	**Magnification**	**Sample Code**		
		**GR/PLA-HP00**	**GR/PLA-HP45**	**GR/PLA-HP90**
Digital image		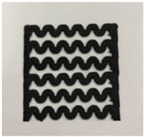	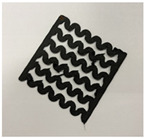	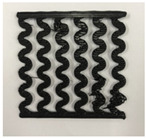
Surface	×6.5	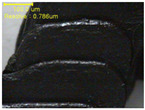	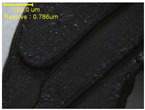	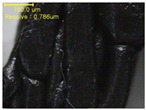
Lateral		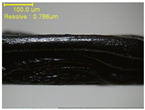	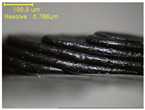	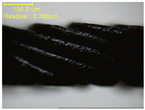

**Table 5 polymers-12-02955-t005:** Morphology of CFDM 3D-printed horseshoe pattern with different 3D printing directions.

**Measured Part**	**Magnification**	**Sample Code**		
		**PLA-HP00**	**PLA-HP45**	**PLA-HP90**
Surface	×300	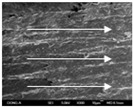	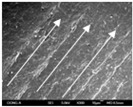	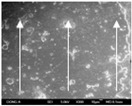
	×1000	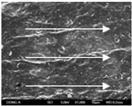	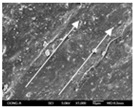	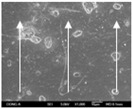
	×10,000	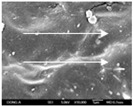	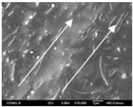	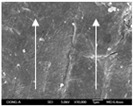
	×15,000	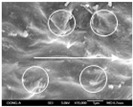	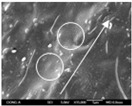	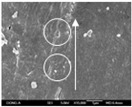
**Measured Part**	**Magnification**	**Sample Code**		
		**GR/PLA-HP00**	**GR/PLA-HP45**	**GR/PLA-HP90**
Lateral	×300	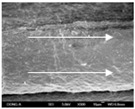	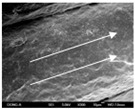	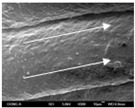
	×1000	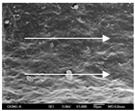	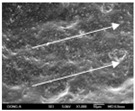	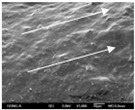
	×10,000	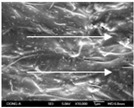	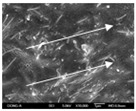	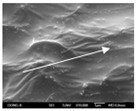
	×15,000	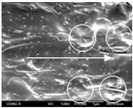	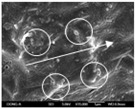	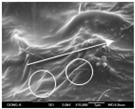

**Table 6 polymers-12-02955-t006:** IR images of PLA CFDM 3D-printed horseshoe pattern with different 3D printing directions.

**PLA-HP00**	**Applied Voltage (V)**
**5**	**10**	**15**	**20**	**25**	**30**
IR image	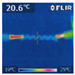	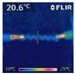	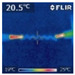	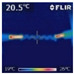	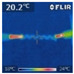	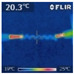
Mean temperature (°C)	20.5 ± 0.2	20.5 ± 0.2	20.3 ± 0.3	20.4 ± 0.1	20.3 ± 0.3	20.4 ± 0.1
Current (A)	0.00 ± 0.00	0.00 ± 0.00	0.00 ± 0.00	0.00 ± 0.00	0.00 ± 0.00	0.00 ± 0.00
**PLA-HP45**	**Applied Voltage (V)**
**5**	**10**	**15**	**20**	**25**	**30**
IR image	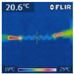	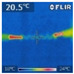	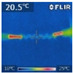	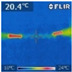	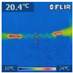	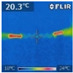
Mean temperature (°C)	20.5 ± 0.2	20.3 ± 0.3	20.4 ± 0.1	20.2 ± 0.2	20.3 ± 0.2	20.3 ± 0.1
Current (A)	0.00 ± 0.00	0.00 ± 0.00	0.00 ± 0.00	0.00 ± 0.00	0.00 ± 0.00	0.00 ± 0.00
**PLA-HP90**	**Applied Voltage (V)**
**5**	**10**	**15**	**20**	**25**	**30**
IR image	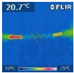	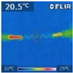	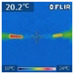	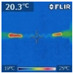	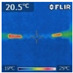	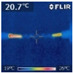
Mean temperature (°C)	20.5 ± 0.2	20.4 ± 0.1	20.2 ± 0.2	20.2 ± 0.2	20.6 ± 0.2	20.6 ± 0.2
Current (A)	0.00 ± 0.00	0.00 ± 0.00	0.00 ± 0.00	0.00 ± 0.00	0.00 ± 0.00	0.00 ± 0.00

**Table 7 polymers-12-02955-t007:** IR images of GR/PLA CFDM 3D-printed horseshoe pattern with different 3D printing directions.

**GR/PLA-HP00**	**Applied Voltage (V)**
**5**	**10**	**15**	**20**	**25**	**30**
IR image	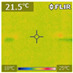	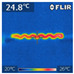	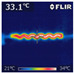	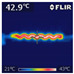	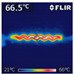	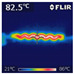
Mean temperature (°C)	21.9 ± 0.5	25.5 ± 0.9	33.5 ± 3.6	45.2 ± 5.7	63.3 ± 6.8	83.6 ± 9.2
Current (A)	0.00 ± 0.00	0.01 ± 0.00	0.02 ± 0.01	0.03 ± 0.01	0.03 ± 0.00	0.04 ± 0.01
**GR/PLA-HP45**	**Applied Voltage (V)**
**5**	**10**	**15**	**20**	**25**	**30**
IR image	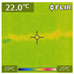	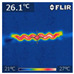	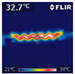	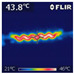	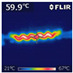	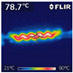
Mean temperature (°C)	21.5 ± 0.4	25.3 ± 1.3	32.3 ± 1.4	45.2 ± 2.9	62.5 ± 5.3	80.6 ± 5.8
Current (A)	0.00 ± 0.00	0.01 ± 0.00	0.02 ± 0.00	0.02 ± 0.00	0.03 ± 0.00	0.04 ± 0.00
**GR/PLA-HP90**	**Applied Voltage (V)**
**5**	**10**	**15**	**20**	**25**	**30**
IR image	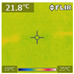	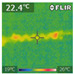	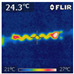	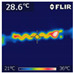	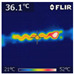	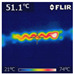
Mean temperature (°C)	21.3 ± 0.4	22.6 ± 0.3	25.6 ± 1.3	30.8 ± 3.2	40.1 ± 5.0	52.5 ± 3.1
Current (A)	0.00 ± 0.00	0.00 ± 0.00	0.01 ± 0.00	0.02 ± 0.01	0.02 ± 0.00	0.03 ± 0.00

**Table 8 polymers-12-02955-t008:** IR images of GR/PLA CFDM 3D-printed horseshoe pattern cotton fabric with different 3D printing directions.

GR/PLA/Co-HP00	Applied Voltage (V)
5	10	15	20	25	30
IR image	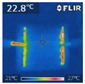	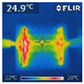	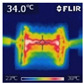	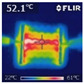	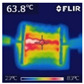	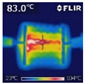
Mean temperature (°C)	22.6 ± 0.5	25.1 ± 1.3	33.5 ± 3.1	51.9 ± 4.7	64.4 ± 6.6	82.6 ± 8.7
Current (A)	0.01 ± 0.00	0.03 ± 0.01	0.06 ± 0.01	0.10 ± 0.02	0.12 ± 0.02	0.15 ± 0.02
**GR/PLA/Co-HP45**	**Applied Voltage (V)**
**5**	**10**	**15**	**20**	**25**	**30**
IR image	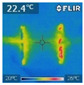	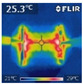	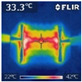	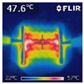	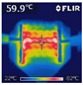	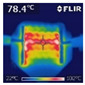
Mean temperature (°C)	22.3 ± 0.2	24.9 ± 0.8	32.6 ± 2.4	48.9 ± 3.9	61.5 ± 6.3	79.6 ± 7.8
Current (A)	0.01 ± 0.00	0.03 ± 0.00	0.06 ± 0.01	0.11 ± 0.02	0.13 ± 0.02	0.15 ± 0.02
**GR/PLA/Co-HP90**	**Applied Voltage (V)**
**5**	**10**	**15**	**20**	**25**	**30**
IR image	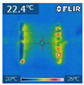	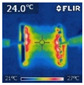	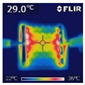	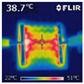	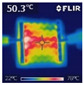	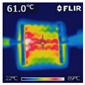
Mean temperature (°C)	22.2 ± 0.4	23.8 ± 0.6	28.5 ± 1.2	39.8 ± 2.9	50.1 ± 4.1	61.5 ± 3.7
Current (A)	0.01 ± 0.00	0.03 ± 0.01	0.05 ± 0.01	0.08 ± 0.01	0.12 ± 0.01	0.15 ± 0.01

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
