# Peer review of "Characterization of Electrical Heating Performance of CFDM 3D-Printed Graphene/Polylactic Acid (PLA) Horseshoe Pattern with Different 3D Printing Directions"

_polymers, 2020, doi:10.3390/polym12122955_

Round 1

Reviewer 1 Report

The authors reported the difference of the structures and electrical heating performance of graphene/PLA horseshoe patterns prepared by CFDM 3D printing with different printing directions. This work provides detailed characterizations and measurements on the structures and heating performances, which will be helpful for the development of graphene-based composite materials. It can be accepted for publication before following issues are addressed.

  1. From the SEM images in Table 5, it is difficult to judge the stacking directions of the graphene layers, which may not be a solid data to further prove the difference of heating performances between graphene/PLA patterns with different directions.
  2. In Figure 3, I don’t think the GR/PLA-HP90 is highly different from GR/PLA-HP45 and GR/PLA-HP00, because all the peaks of GR and PLA are suppressed in GR/PLA-HP90. It is high possibility that the XRD measurements were not conducted at the same conditions, especially the X-ray intensity.
  3. In my opinion, the resistivity between GR/PLA-HP90, GR/PLA-HP45, and GR/PLA-HP00 is not from the differences of graphene layer directions, but from the adhesion or gaps between each printing lines. From the 30V data column in Table 8, it can be clearly known that the current becomes same because the high temperature induces fusion of the printing lines, which supports this opinion.

Author Response

Attach the files.

Reviewer 2 Report

I am glad that I had an oportunity to reviwe attached paper. in my opinion it is quite good  and interesting. The investigation methods are good selected and  and well reported. The obtained results can find very interestig application. Here you have some comments to the article:

1. Abstract, page 1, line 26, the sentence:" When 30V was applied to three GR/PLA-HP samples.(...)the surface temperatures increased??? in the order of ...."
should be rewriten - beacause temperature decreased!

2. Abstract, page 1, line 30: What does "electrical properties" mean to authors?
As for me, good electrical propertis means that material exhibits high conductivity ...and low electrical resistance. If comes about elactrical heating properties they are connected to high material's resitance. So I think it should be changed/rewriten  to make it clear. 
3. Introduction, page 1, line 36: "...industrial fields, including in the production...." - "in" should be removed.

4. Introduction, page line 42: the sentence " recently, a method to solve this problem was developed....." In my opinion it should be replaced by the sentence: " Recently, to solve the problem, the method was developed....."
5. Characterization, page 5, lines 126-130: there is no units (m, mm, ...)  mentioned in the description!
6. Characterization, page 5: Authors should provide information about measurement of electrical current for exemple about the accurancy. In Tables 6, 7, 8 we have information that deviation of current value is about 0.00?? How it can be? What was the sensivity of the mesurement method?
 7. page 12, Figure 6: How can you explain significant deviation in the current value when  20 V was applied to the samples? In the same figure, for sample PLA-HP indicators shoud be changed or maybe these data should be removed because pure PLA is not a conductive material.

But inspite of some my suggestions, doubts and comments this quite good work.

Author Response

Attach the files.

Round 2

Reviewer 1 Report

All the comments have been addressed.